# Assessing Genomic Mutations in SARS-CoV-2: Potential Resistance to Antiviral Drugs in Viral Populations from Untreated COVID-19 Patients

**DOI:** 10.3390/microorganisms12010002

**Published:** 2023-12-19

**Authors:** Daniele Lombardo, Cristina Musolino, Valeria Chines, Giuseppe Caminiti, Claudia Palermo, Irene Cacciola, Giuseppina Raffa, Teresa Pollicino

**Affiliations:** 1Department of Clinical and Experimental Medicine, University Hospital of Messina, 98124 Messina, Italy; daniele.lombardo@unime.it (D.L.); valeria.chines@unime.it (V.C.); giuseppe.caminiti@unime.it (G.C.); claudia_palermo@icloud.com (C.P.); irene.cacciola@unime.it (I.C.); giuseppina.raffa@unime.it (G.R.); 2Department of Human Pathology, University Hospital of Messina, 98124 Messina, Italy; cristina.musolino@unime.it

**Keywords:** SARS-CoV-2, COVID-19, coronaviruses, antiviral drugs, antiviral therapy, antiviral drug resistance, new generation sequencing, viral genomic mutation, natural variants

## Abstract

Naturally occurring SARS-CoV-2 variants mutated in genomic regions targeted by antiviral drugs have not been extensively studied. This study investigated the potential of the RNA-dependent RNA polymerase (RdRp) complex subunits and non-structural protein (Nsp)5 of severe acute respiratory syndrome coronavirus-2 (SARS-CoV-2) to accumulate natural mutations that could affect the efficacy of antiviral drugs. To this aim, SARS-CoV-2 genomic sequences isolated from 4155 drug-naive individuals from southern Italy were analyzed using the Illumina MiSeq platform. Sequencing of the 4155 samples showed the following viral variant distribution: 71.2% Delta, 22.2% Omicron, and 6.4% Alpha. In the Nsp12 sequences, we found 84 amino acid substitutions. The most common one was P323L, detected in 3777/4155 (91%) samples, with 2906/3777 (69.9%) also showing the G671S substitution in combination. Additionally, we identified 28, 14, and 24 different amino acid substitutions in the Nsp5, Nsp7, and Nsp8 genomic regions, respectively. Of note, the V186F and A191V substitutions, affecting residues adjacent to the active site of Nsp5 (the target of the antiviral drug Paxlovid), were found in 157/4155 (3.8%) and 3/4155 (0.07%) samples, respectively. In conclusion, the RdRp complex subunits and the Nsp5 genomic region exhibit susceptibility to accumulating natural mutations. This susceptibility poses a potential risk to the efficacy of antiviral drugs, as these mutations may compromise the drug ability to inhibit viral replication

## 1. Introduction

Ongoing and relentless, the Coronavirus Disease 2019 (COVID-19) pandemic continues to inflict disruptions, confirming its status as the most profound and devastating global health challenge of the past century. Severe acute respiratory syndrome coronavirus 2 (SARS-CoV-2) is the pathogen responsible for COVID-19. It is an enveloped, positive-sense, single-stranded RNA virus with a genome size of approximately 30,000 base pairs [1]. While SARS-CoV-2 primarily infects the respiratory tract and lungs, often referred to as the “primary site of infection” [2], evidence suggests that it can also spread to other organs [3].

Upon infecting a host cell, SARS-CoV-2 utilizes a 5′-capped and 3′-polyadenylated RNA genome to directly interact with the host protein synthesis machinery. In the initial phase of genomic RNA translation, host ribosomes, translation factors, and RNA-binding proteins produce two polyproteins encoded by open reading frame (ORF)1a and ORF1ab. ORF1ab encompasses 16 non-structural proteins (Nsps) that undergo proteolytic processing to generate enzymes essential for viral RNA synthesis. Specifically, Nsp2 to Nsp11 contribute to the formation of the viral replication-transcription complex (RTC), while Nsp12 to Nsp16 perform core enzymatic functions crucial for viral RNA synthesis, RNA proofreading, and RNA modification. Beyond the Nsps, SARS-CoV-2’s genetic code includes four structural proteins (nucleocapsid (N), envelope (E), membrane (M), and spike (S)) and eight accessory proteins (ORF3a–ORF9b), expressed from a nested set of 5′ and 3′ co-terminal subgenomic mRNAs [1].

Nsps, being highly conserved proteins integral to key processes in the SARS-CoV-2 life cycle, represent prime targets for potential antiviral drug development [4]. However, the rapid emergence of SARS-CoV-2 variants poses a significant challenge to the clinical application of currently available therapeutics.

The development of safe and effective antiviral drugs has been a key issue since the outbreak of the global pandemic SARS-CoV-2 [5,6]. Remdesivir, also known as Veklury, was the first drug among those tested for the treatment of SARS-CoV-2 infected patients to receive approval for treating COVID-19 by both the Food and Drug Administration (FDA) [7] and the European Medicines Agency (EMA) [8]. Remdesivir targets Nsp12, which is the catalytic subunit of the RNA-dependent RNA polymerase (RdRp) complex [5,9].

The replication and transcription of the SARS-CoV-2 genome rely on the RdRp complex, which consists of the catalytic subunit Nsp12, along with the two accessory subunits, Nsp7 and Nsp8 [10]. The Nsp12 subunit, spanning amino acids S367 to F920, comprises three distinct domains: An N-terminal nidovirus RdRp-associated nucleotidyltransferase (NiRAN) domain, an interface domain, and a C-terminal RdRp domain [11]. The RdRp structure resembles a right hand and is composed of three subdomains: (a) the fingers subdomain, comprising residues L366 to A581 and K621 to G679, (b) the palm subdomain, comprising residues T582 to P620 and T680 to Q815, and (c) the thumb subdomain, comprising residues H816 to F920. Subunits Nsp7 and Nsp8 bind to the thumb subdomain, while an additional copy of Nsp8 binds to the fingers subdomain [11]. Remdesivir effectively eludes the SARS-CoV-2 RdRp proofreading activity by allowing RNA polymerase elongation to proceed beyond the site of incorporation by adding three additional nucleotides, with consequent accumulation of incorrect nucleotides and consequent interruption of viral replication [9,12,13]. Another drug recently approved by the United Kingdom (UK) Medicines Regulatory Agency [14] and the FDA [15] for the treatment of patients with COVID-19 is Molnupiravir (or MK 4482 or Lagevrio), which also targets the RdRp of SARS-CoV-2. In contrast to Remdesivir, which is administered by infusion, Molnupiravir is orally bioavailable. The available data suggest that Molnupiravir functions as a mutagenic agent, leading to an “error catastrophe” [16] (pp. 740) during viral replication [17]. However, the EMA has recently withdrawn this drug from the market for treating COVID-19 in adult patients due to insufficient or inadequate evidence of its efficacy [18]. A third antiviral drug for the treatment of COVID-19 has been introduced with the approval of Paxlovid, which is a combination of Nirmatrelvir (PF-07321332) and ritonavir [19,20]. Paxlovid is orally bioavailable and can be administered at the initial signs of SARS-CoV-2 infection, potentially assisting patients in avoiding severe illness, hospitalization, and mortality. Unlike Remdesivir, Paxlovid is specifically designed to inhibit the activity of Nsp5, which is also known as the main protease (M^pro^) or 3-chymotrypsin-like cysteine protease (3CL^pro^) of SARS-CoV-2. The structure of Nsp5 includes a chymotrypsin-like domain (domain I, residues 8–101), a 3C protease-like domain (domain II, residues 102–184), and a globular domain (domain III, residues 201–303) responsible for the dimerization of two different protomers of Nsp5 [21,22]. The dimerization of two protomers results in the formation of a catalytic site for each protomer, where C145 and H41 represent the catalytic residues [21,22]. Nsp5 is responsible for the maturation of non-structural proteins, including those of the RdRp complex [23]. Therefore, Paxlovid inhibits viral replication acting upstream the replication complex [24].

Considering the adaptive potential of SARS-CoV-2 and the ongoing selection of viral variants with some degree of resistance to vaccines, it is conceivable that natural selection of mutant viruses could also jeopardize the efficacy of newly developed antiviral agents.

Typically, mutations conferring drug resistance emerge as a direct consequence of widespread drug usage, where pharmacological selective pressure drives the rapid proliferation of resistant strains [25]. However, pre-existing genetic variations, particularly recurrent ones, at drug-binding sites within bacterial or viral populations offer valuable insights into the potential for future drug resistance development. The presence of these mutations in a drug-naive population suggests that the pathogen harbors an inherent tolerance for modifications at these specific positions. Consequently, these mutations can be rapidly “rediscovered” by the pathogen once drug pressure is exerted [25]. Various molecular factors contribute to the establishment of viral quasi-species shortly after infection. These factors include the rapid rate of viral replication, characterized by a high mutation rate (estimated at ∼1.12 × 10^−3^ nt^−1^ year^−1^ for SARS-CoV-2) [26], large viral load, viral fitness, and ample replication space [27]. Under appropriate selective pressure, minor drug-resistant variants within the quasi-species can evolve into dominant susceptible “wild-type” viruses [27]. The primary mechanism underlying acquired drug resistance in viruses involves random mutations within viral genes, which alter the drug’s binding affinity to its protein target, rendering the drug ineffective. Consequently, the emergence of drug-resistant viral strains poses a persistent challenge, leading to treatment failures and increased morbidity and mortality [28].

The identification and prevention of potential antiviral-resistant SARS-CoV-2 variants are critical components of infection surveillance. Notably, two separate studies from different countries have reported non-synonymous mutations in Nsp12 [29,30]. Additionally, a case report has provided evidence linking clinical treatment failure with Remdesivir to the D484Y Nsp12 substitution [31]. Recent in vitro studies have further indicated that amino acid substitutions F480L, V557L, and E802D in Nsp12 might potentially reduce the sensitivity of SARS-CoV-2 to Remdesivir [32]. Concerning Paxlovid, a recent study has demonstrated that amino acid substitutions E166V and S144E in 3CL^pro^ can significantly impair the inhibitory activity of this antiviral drug on viral replication or protease activity, reducing it by more than 100-fold [33]. Despite the importance of this clinical concern, it has received limited attention in research, and to our understanding, no studies have yet explored the selection of naturally occurring variants containing mutations within the genomic regions of SARS-CoV-2 that are targeted by antiviral drugs in populations from specific geographic locations.

The principal objective of this study was to examine the occurrence of SARS-CoV-2 variants and analyze the genetic diversity of the RdRp complex subunits and Nsp5 within SARS-CoV-2 genomic sequences obtained from a large group of individuals residing in southern Italy, all of whom had not undergone antiviral treatment.

## 2. Materials and Methods

### 2.1. Samples

From April 2021 to October 2022, a total of 4155 nasopharyngeal swabs that were positive for SARS-CoV-2 by PCR diagnostic testing at various diagnostic centers in eastern Sicily were subjected to SARS-CoV-2 variant analysis at the Molecular Diagnostic Laboratory of the Unit of Advanced Diagnostic Laboratories, University Hospital of Messina, Italy. This laboratory serves as “secondary research center” [34] and is one of the Italian regional reference laboratories for the identification of SARS-CoV-2 emerging variants in Italy.

The biosamples utilized in this study underwent anonymization via key coding prior to their arrival at our laboratory. In this procedure, a unique identifier was assigned to each sample, guaranteeing the absence of any retained personal information. This process ensured that the biosamples could be analyzed without compromising the privacy of the individuals from whom they were collected. This study was conducted in accordance with the principles of the 1964 Declaration of Helsinki, and informed consent was waived in accordance with Italian governmental regulations on observational retrospective studies [35].

### 2.2. Viral Nucleic Acid Extraction

Viral RNA extraction was carried out utilizing the Maxwell RSC 48 automated nucleic acid purification platform (Promega Corporation, Madison, WI, USA) with the use of the Maxwell RSC Viral TNA kit (Promega Corporation), following the manufacturer’s prescribed protocols and instructions. Briefly, 200 µL of viral transport medium (VTM) from nasopharyngeal swab tubes was treated with Lysis Solution containing lysis buffer and proteinase K. The solution was vortexed for 10 s and incubated at 56 °C in a heat block for 10 min. During incubation, the reagent cartridges and elution tubes were placed in the instrument. Subsequently, the sample lysates were transferred into each extraction well of the cartridges. Each sample was then eluted in 50 µL of nuclease-free water.

### 2.3. Sequencing of SARS-CoV-2 Isolates

Viral RNA reverse transcription, cDNA amplification, and library construction were performed using the QIAseq DIRECT SARS-CoV-2 kit (Qiagen, Hilden, Germany) according to the manufacturer’s instructions and sequenced on the MiSeq platform (Illumina, San Diego, CA, USA) with the number of sequencing cycles set to 149 bp and the pair-end sequencing option. Briefly, each extracted RNA served as a template for random-primed cDNA synthesis (1 cycle of 10 min at 25 °C, 1 cycle of 50 min at 42 °C, and 1 cycle of 5 min at 85 °C), generating a comprehensive library of SARS-CoV-2 complementary DNA (cDNA). Subsequently, the resulting cDNA was amplified using a PCR master mix containing multiplexed SARS-CoV-2-specific primer pools (Qiagen), yielding two enriched pools of approximately 225–275 bp overlapping amplicons for each sample. Then, for each sample, the two pools were combined into a single tube and purified using QIAseq magnetic beads (Qiagen) to eliminate primers, enzymes, and other contaminants. The purified library was quantified using a fluorometer to determine the DNA concentration in the library. Each enriched sample was then amplified and indexed using the QIAseq unique dual indexes (UDIs, Qiagen) following the manufacturer’s instructions (1 cycle of 2 min at 98 °C followed by 4 cycles of 20 s at 98 °C and 5 min at 63 °C, followed by 20 cycles of 20 s at 98 °C and 3 min at 63 °C), and further purified using QIAseq magnetic beads (Qiagen). The purified libraries were pooled and sequenced on the MiSeq platform (Illumina).

### 2.4. Genetic Analysis

The complete SARS-CoV-2 genomes were reconstructed using the SARS-CoV-2 REconstruction of COronaVirus gEnomes & Rapid analYsis (RECoVERY) pipeline [36]. Briefly, the pipeline provides the following steps: reads quality analysis and trimming, subtraction of human sequences, reads alignment and mapping against the SARS-CoV-2 reference sequence, variant calling, consensus sequence calling, ORFs identification, and variant annotation. The raw reads, obtained from sequencing, were analyzed using the trimmomatic program (v0.39) to eliminate those of low quality and with a length of less than 30 base pairs. Subsequently, the Bowtie2 program (v2.5.2) was used to map the reads onto the reference human genome, downloaded from the “the genome reference consortium” database (https://www.ncbi.nlm.nih.gov/grc, accessed on 23 February 2021), to eliminate human genomic sequences. The remaining sequences were then mapped to the SARS-CoV-2 reference genome (NC_045512.2) using the Bowtie2 program and the resulting alignments were analyzed using the iVar consensus caller (v1.4.2) program with the following options: (1) minimum quality score threshold to count base 20, (2) minimum frequency threshold 0.6, and (3) minimum depth to call consensus 30. The coverage analysis was performed using Qualimap 2 (v2.3) program. The ORFs identification and annotation was performed using BLASTn (https://blast.ncbi.nlm.nih.gov/Blast.cgi?PROGRAM=blastn&BLAST_SPEC=GeoBlast&PAGE_TYPE=BlastSearch, accessed on 7 November 2022), setting nucleotide identity >99% as a requirement. The parameters set for the alignment were: one as maximum number of hits, 80% identity cut-off, 80% minimum query coverage per High-scoring Segment Pair (HSP) as indicated in the reference’s protocol [36]. Finally, the variant calling was performed using iVar variant caller (v1.4.2) on the file resulted from the mapping of the reads to the reference of SARS-CoV-2 with minimum quality (Default: 20) and minimum frequency (modified: 0.3). The SnpEff tool (v5.2) was eventually used for variants annotation. The analysis of SARS-CoV-2 mutation within the Nsp5, 7, 8, and 12 genes was performed using the Aliview program (v1.27). COVID Lineage analysis was performed using Dragen COVID Lineage v3.5.2 (Illumina) on BaseSpace Sequence Hub (Illumina) following a standard protocol.

The 4155 SARS-CoV-2 sequences obtained in this study are openly available on GISAID portal (https://gisaid.org, accessed on 7 December 2023) under the GISAID Identifier: EPI_SET_231207zf, digital object identifier (DOI): https://doi.org/10.55876/gis8.231207zf, accessed on 7 December 2023. All sequences in this dataset are compared relative to hCoV-19/Wuhan/WIV04/2019 (WIV04), the official reference sequence employed by GISAID (EPI_ISL_402124). All the sequences of the dataset were deposited from April 2021 to October 2022 (Appendix A).

### 2.5. Phylogenetic Analysis

To elucidate the phylogenetic relationships among the SARS-CoV-2 consensus sequences, we employed Molecular Evolutionary Genetics Analysis X (MEGAX) v11.0.13 [37]. Initially, we performed Multiple Sequence Alignment (MSA) using the Muscle algorithm to realign the consensus sequences of the Nsp5, 7, 8, and 12 genes. This realignment ensured that corresponding positions in the sequences were appropriately aligned, enabling meaningful comparisons. Subsequently, the newly aligned sequences were used to construct a phylogenetic tree with default parameters, utilizing the maximum likelihood method and the Tamura–Nei model, a statistical model used to estimate the evolutionary relationships between samples based on their DNA sequences. The model accounts for varying substitution rates among nucleotide pairs and estimates the transition–transversion bias parameter. It is used for datasets that are suspected to have heterogeneous substitution rates among sites or have a high guanine and cytosine (G + C) content [38]. To enhance the robustness of the analysis, a bootstrap consensus tree was generated from 500 replicates, which serves as a representation of the virus’s evolutionary history [39]. For visualizing the resulting tree, we employed the online tool Interactive Tree of Life (iTOL) (https://itol.embl.de, accessed on 29 November 2022) [40]. Additionally, all instances of N-bases were substituted with the corresponding nucleotides from the reference genome SARS-CoV-2 NC_045512.2.

## 3. Results

### 3.1. COVID Lineage Analysis

Sequencing analysis of the 4155 samples revealed that 2957 samples (71.2%) contained viral genomes of the B.1.617.2 (Delta) variant, which had different parent lineages. In addition, 924 samples (22.2%) contained viral genome of the B.1.1.529 (Omicron) variant, and 250 samples (6.4%) were of the B.1.1.7 (Alpha) variant, whereas the remaining 24 samples (0.6%) belonged to lesser recurring variants in southern Italy (eastern Sicily) (Figure 1).

In April 2021, neither the Delta variant nor its parent lineages were detected, and the prevalence of the Alpha variant accounted for 88.8% of the cases. However, by the first decade of July 2021, the Delta variant was identified in 85.5% of the samples. In August 2021, it rapidly replaced the Alpha variant, becoming the primary strain in over 95% of cases, a figure that rose to 100% by October and November 2021. In December 2021, the Omicron variant appeared in 17% of cases, quickly overtaking Delta and accounting for 77.1% of cases in January 2022 and 93.5% of the cases in February 2022. Since March 2022, the Omicron variant and its parent lineage have consistently accounted for 100% of the observed cases (Table 1, Figure 2).

### 3.2. Evaluation of the RdRp Complex Variability

#### 3.2.1. Catalytic Subunit

We found the A46S and R197Q substitutions within the NiRAN domain (Figure 3), which were present in 17 (0.4%) and 104 (2.5%) out of 4155 studied samples, respectively. All isolated SARS-CoV-2 containing these substitutions belonged to the Delta variant. Additionally, in the NiRAN domain we also found the N215D substitution, which instead was associated with the Omicron variant. Within the interface domain, which connects the NiRAN domain to the polymerase domain [11], we identified the P323L substitution. This substitution was present in 3777 out of the 4155 samples (91%). Among the 3777 samples carrying the P323L substitution, 219 (5.8%) were associated with the Alpha variant, 2615 (69.2%) with the Delta variant, 921 (24.4%) with the Omicron variant, and the remaining 22 samples (0.6%) with other variants (1 with B.1.351, 11 with the P.1, 2 with the B.1.525, 1 with the P.3, and 5 with the C.37). The polymerase domain is divided into three subdomains: the fingers subdomain, the palm subdomain, and the thumb subdomain [11]. Within the fingers subdomain, we observed the following amino acid (aa) substitutions: M463I (in 25 out of 4155 samples, 0.6%), V472L (in 34 out of 4155 samples, 0.8%), and A656V (in 20 out of 4155 samples, 0.5%). All of these substitutions were associated with the Delta variant. Additionally, the G671S substitution was detected in 2906 out of 4155 samples (69.9%). Among these, 2904 samples (99.05%) were associated with the Delta variant, while the remaining 2 samples (0.05%) were associated with BA.5, a sub-variant of the Omicron variant. Concerning the palm subdomain, we observed the following amino acid substitutions: A716V (in 16 out of 4155 samples, 0.4%), associated with the Alpha variant, and D804N (in 26 out of 4155 samples, 0.6%), associated with the Delta variant. Finally, within the thumb subdomain we identified several amino acid substitutions and a specific deletion. In particular, we detected the L838I substitution in 457/4155 samples (11%), the Q822H substitution in 140/4155 samples (3.4%), the D824Y substitution in 21/4155 samples (0.5%), and the deletion of amino acid T908 in 26/4155 samples (0.6%). All these mutations were associated with the Delta variant. The variability of mutations observed across different SARS-CoV-2 variants was particularly striking, with some mutations being highly prevalent across multiple variants, suggesting their potential impact on viral fitness and transmission. Conversely, mutations specific to certain variants indicate their potential role in unique adaptations. All the mutations in Nsp12 sequences that we detected in this study are reported in Appendix A.

#### 3.2.2. Accessory Subunits

The molecular analysis of non-structural proteins Nsp7 and Nsp8 of SARS-CoV-2 revealed the presence of a total of 14 and 24 amino acid substitutions, respectively, though with low frequency (Figure 3). Specifically, in the case of Nsp7, the L56F substitution was identified in 16/4155 samples (0.4%), while the remaining 13 substitutions showed lower prevalence levels (Appendix A). In the context of Nsp8, two substitutions, namely M55I and T148I, were detected in 34 and 33 out of 4155 samples, respectively, both accounting for a prevalence of 0.8%. The remaining 22 amino acid substitutions exhibited reduced prevalence rates, as described in Appendix A.

### 3.3. Evaluation of Nsp5 Variability

The molecular analysis of SARS-CoV-2 Nsp5 revealed the presence of 28 amino acid substitutions distributed across the three domains of the protein (Appendix A). Notably, among these substitutions, three exhibited higher prevalence rates (Figure 3). Indeed, the K90R substitution was identified in 118/4155 (2.8%), the P132H substitution in 648/4155 cases (15.6%), and the V186F substitution in 157/4155 cases (3.8%). It is noteworthy that all samples containing the P132H substitution were associated with the Omicron and parent lineages, while those with the V186F substitution were linked to the Delta and parent lineages. In contrast, the K90R substitution was less lineage-specific, as it was present in both the Omicron and Delta lineages, as well as in other lineages (Appendix A). Additionally, a mutation of potential significance, the A191V substitution, was detected in 3/4155 samples (0.07%). Importantly, this mutation affects an amino acid residue adjacent to the catalytic site of Nsp5 [19].

### 3.4. Phylogenetic Analysis

Phylogenetic trees are represented in a circular cladogram format due to the extensive number of sequences analyzed. In this representation, the SARS-CoV-2 NC_045512.2 sequence, serving as a reference for each of the examined genes, is positioned at the center of the cladogram. The length of each branch indicates the evolutionary distance or genetic divergence between sequences, with longer branches indicating greater dissimilarity. Sequences that share a closer genetic relationship are positioned closer to one another on the circular cladogram, while more distantly related sequences are placed farther along the circle. When comparing the phylogenetic distances of Nsp7 and Nsp8 sequences to those of Nsp12 and Nsp5, it is evident that the variation in Nsp7 and Nsp8 sequences is less pronounced. Consequently, Nsp7 and Nsp8 sequences from lineages such as B.1.1.7, B.1.617.2, and their parent lineages cluster closely together, as do sequences from B.1.529 and its parent lineages. The results of the phylogenetic analysis are illustrated in Figure 4a–d.

## 4. Discussion

The rapid and global spread of SARS-CoV-2 infection has led to the rapid selection and emergence of different viral variants and lineages harboring amino acid substitutions that can affect the efficacy of antiviral therapy [29,30,31]. These substitutions may be naturally selected, such as E802D—the only spontaneous substitution known to decrease susceptibility to Remdesivir in vitro [32]—or emerge in response to antiviral therapy, such as D484Y, the only amino acid substitution detected after Remdesivir administration and associated with treatment failure [31]. Interestingly, recent articles have suggested the influence of natural products produced by the commensal microbiota as an environmental selection pressure contributing to the emergence of new variants of SARS-CoV-2 [41,42,43].

In this study, we identified several mutations in SARS-CoV-2 genomic regions that are targeted by antiviral drugs. Analysis of Nsp12 sequences identified 84 amino acid substitutions, of which P323L was the most common, detected in 3777/4155 (91%) of the samples. This result is consistent with prevalence data from North America, India, and Europe [44,45]. Amino acid P323 is situated within the interface domain spanning residues A250 to R365 [10], which closely interacts with Nsp8 to regulate RNA synthesis [44,45]. Substitution of a non-polar amino acid, like proline, with a hydrophobic residue, like leucine, increases the hydrophobic interaction between Nsp12 and Nsp8, potentially further enhancing the processivity of RNA synthesis by the RdRp complex. Thus, the P323L substitution might have contributed to the epidemiological spread of the Omicron variant [29,46]. However, the high prevalence of this variant would not necessarily have a negative impact on antiviral therapy, also taking into account that a recent study suggests that the P323L substitution increases the affinity of Nsp12 for Remdesivir, resulting in greater treatment efficacy [47].

Moreover, within the Nsp12 gene of SARS-CoV-2, G671S was another amino acid substitution that showed a high prevalence (69.9%) in isolated SARS-CoV-2 genomes. This substitution always occurred in combination with P323L. The P323L/G671S combination was found in Delta variants and in two samples belonging to the Omicron BA.5 sub-variant. Interestingly, it has been shown [48] that the G671S substitution likely enhances the activity and stability of the RdRp complex, thereby promoting SARS-CoV-2 replication. This could potentially account for the increased prevalence observed in our study. Currently, it remains unknown whether the association of P323L and/or P323L/G671S might confer any level of resistance to antiviral drugs. Examination of the two pivotal co-factors, Nsp7 and Nsp8, within the RdRp complex has unveiled an extremely limited genetic diversity. Notably, among the relatively few amino acid substitutions detected, the M55I substitution in Nsp8 was the one with the highest prevalence (0.8% of the samples, Appendix A). Given the central roles played by these co-factors in significantly enhancing the efficiency of the polymerization reaction [45,49] and their lower susceptibility to amino acid substitutions compared to Nsp12, it becomes an intriguing prospect to contemplate Nsp7 and Nsp8 as potential focal points for the development of antiviral drugs.

Regarding the main SARS-CoV-2 protease Nsp5, which is the target of the antiviral drug Paxlovid, no amino acid substitutions were identified in the active site of this protein. However, it is worth noting that the V186F and A191V substitutions, affecting residues adjacent to the Nsp5 active site cavity, were found in 3.8% and 0.07% of the 4155 isolated SARS-CoV-2 samples (Appendix A). This highlights the need for ongoing surveillance and research to monitor for the emergence of potential resistance-related mutations.

Phylogenetic analysis of isolated SARS-CoV-2 strains has revealed a fascinating complexity in the evolutionary rates of different genomic regions. In particular, certain regions show increased mutation rates, while others remain highly conserved. In particular, both Nsp7 and Nsp8 sequences from evolutionarily distant lineages tend to cluster together, highlighting their relative conservation compared to Nsp12 and Nsp5 (as shown in Figure 4. This intricate pattern of evolution can pose challenges in the interpretation of phylogenetic trees. Branches associated with higher mutation rates may appear longer, potentially misleading our understanding of evolutionary relationships. In addition, some sequencing reactions may detect genomic variation within an individual patient, which could manifest as additional branches in the tree. However, these branches may not represent independent lineages or distinct transmission events. Nevertheless, despite these complications and potential limitations, phylogenetic trees have played a crucial role in tracking the spread of the virus, identifying potential sources of outbreaks, and improving our understanding of the evolutionary dynamics of SARS-CoV-2.

The results of our study highlight the dynamic genetic diversity of SARS-CoV-2, with variants such as Delta and Omicron exhibiting distinct patterns of amino acid substitutions in key genomic regions and proteins. Although the scope of our study is limited to eastern Sicily, it has shown that even within this restricted area, conserved genomic regions of SARS-CoV-2, such as Nsp5, Nsp7, Nsp8, and Nsp12, are prone to accumulate spontaneous mutations in individuals who have not been exposed to antiviral treatments. Some of these mutations, might compromise the efficacy of antiviral drugs, which are of fundamental importance for patients at risk of severe COVID-19.

While this study provides valuable insights into the evolutionary landscape of SARS-CoV-2, it is not without limitations. Notably, this study did not delve into the functional consequences of the identified mutations, leaving unanswered questions regarding their impact on viral fitness and transmission. To address these limitations and gain a more comprehensive understanding of the evolutionary trajectory of SARS-CoV-2, future research should focus on conducting functional studies to evaluate the impact of observed mutations on viral replication, infectivity, and immune evasion, and investigating the potential of these mutations to diminish the efficacy of existing antiviral therapies.

The relentless evolution of SARS-CoV-2 poses a significant challenge to the development of future effective antiviral therapies. Understanding the mechanisms by which mutations in RdRp complex subunits and Nsp5 impact drug efficacy is crucial for designing new drugs that are less susceptible to viral resistance.

Several strategies can be employed to mitigate the risk of drug resistance and ensure the continued efficacy of antiviral therapies. One approach involves the development of broad-spectrum antiviral drugs that target multiple viral components, reducing the likelihood that a single mutation can render the drug ineffective. Additionally, combination therapies, which utilize two or more antiviral drugs with different targets, can hinder the emergence of resistant strains by making it more difficult for the virus to develop resistance. Furthermore, continuous monitoring of viral evolution is essential for identifying emerging drug-resistant strains and informing the development of new antiviral strategies. Hence, the use of genetic sequence analysis for evaluating SARS-CoV-2 variability is of utmost significance. This ongoing vigilance is essential for a deeper understanding of how the virus undergoes evolutionary changes, adapts to new conditions, and potentially responds to both immune defenses and therapeutic interventions.

## Figures and Tables

**Figure 1 microorganisms-12-00002-f001:**
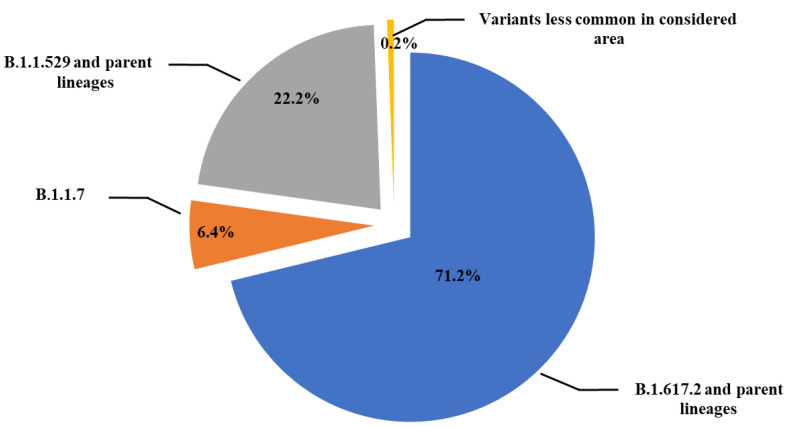
Schematic representation of SARS-CoV-2 lineages prevalence in 4155 samples from subjects with a first positive SARS-CoV-2 PCR test analyzed at the Molecular Diagnostic Laboratory of the Unit of Advanced Diagnostic Laboratories, University Hospital of Messina, Italy, from April 2021 to October 2022.

**Figure 2 microorganisms-12-00002-f002:**
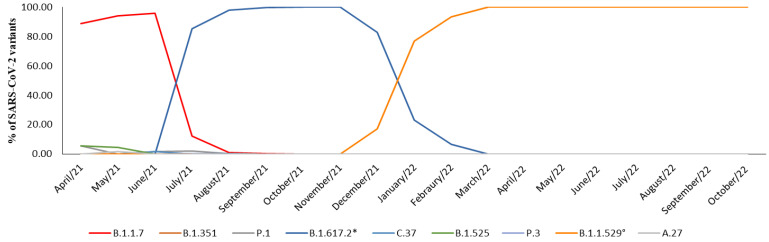
Representation of prevalence expressed in percentage (%) of SARS-CoV-2 variants prevalence between April 2021 and October 2022. * B.1.617.2 and parent lineages; ° B.1.1.529 and parent lineages.

**Figure 3 microorganisms-12-00002-f003:**
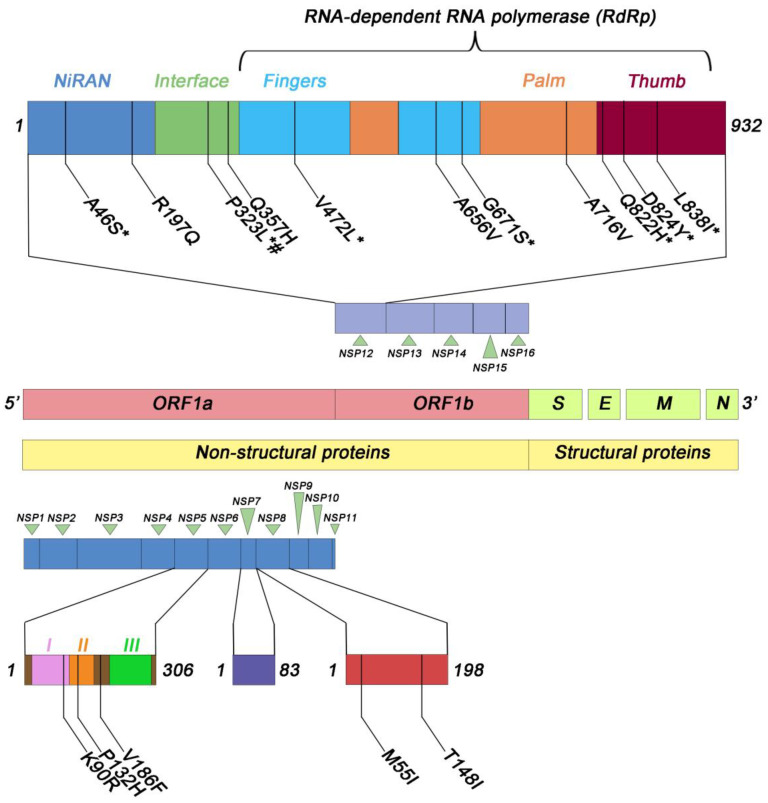
Schematic representation of the most frequent amino acid substitutions identified in Nsp12, 5, 7, and 8 in the samples analyzed. I, II, III represent the three domains of NSP5 protein. # B.1.1.7; * B.1.617.2 and parent lineages.

**Figure 4 microorganisms-12-00002-f004:**
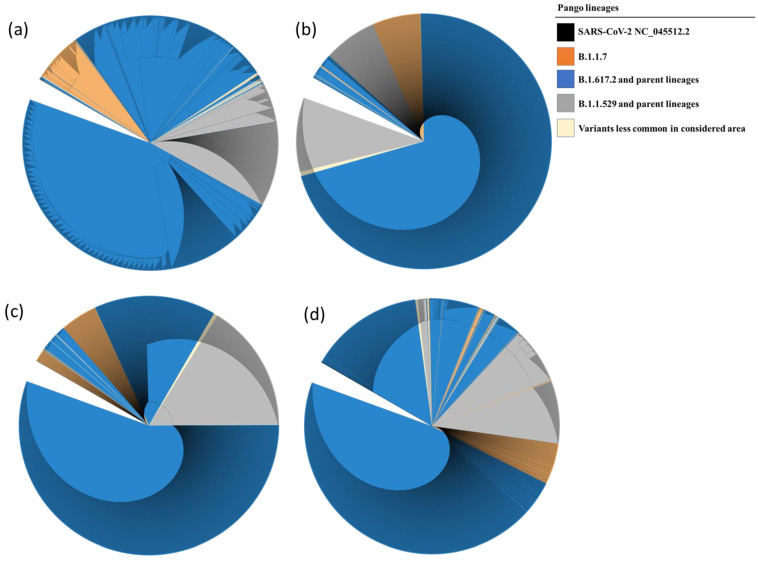
Phylogenetic tree of Nsp12 (**a**), Nsp7 (**b**), Nsp8 (**c**), and Nsp5 (**d**) showing evolutionary relationship between the samples analyzed and the reference sequence. The evolutionary history was inferred by using the Maximum Likelihood method and the Tamura–Nei model. The presence of single nucleotide mutations caused the further subdivision of the clades in different subgroups.

**Table 1 microorganisms-12-00002-t001:** Prevalence of SARS-CoV-2 lineages from April 2021 to October 2022 in samples analyzed.

WHO Label	Alpha	Beta	Gamma	Delta	Lambda	Eta	Theta	Omicron	None
Pango Lineage	B.1.1.7	B.1.351	P.1	B.1.617.2 *	C.37	B.1.525	P.3	B.1.1.529 °	A.27
April/2021	88.8%	-	5.6%	-	-	5.6%	-	-	-
May/2021	94.1%	-	-	-	-	4.4%	-	-	1.5%
June/2021	96%	0.8%	1.6%	-	1.6%	-	-	-	-
July/2021	12.2%	-	2%	85.5%	-	-	0.3%	-	-
August/2021	1.20%	0.1%	0.3%	98%	0.4%	-	-	-	-
September/2021	0.2%	-	-	99.8%	-	-	-	-	-
October/2021	-	-	-	100%	-	-	-	-	-
November/2021	-	-	-	100%	-	-	-	-	-
December/2021	-	-	-	83%	-	-	-	17%	-
January/2022	-	-	-	23.1%	-	-	-	76.9%	-
February/2022	-	-	-	6.5%	-	-	-	93.5%	-
March/2022	-	-	-	-	-	-	-	100%	-
April/2022	-	-	-	-	-	-	-	100%	-
May/2022	-	-	-	-	-	-	-	100%	-
June/2022	-	-	-	-	-	-	-	100%	-
July/2022	-	-	-	-	-	-	-	100%	-
August/2022	-	-	-	-	-	-	-	100%	-
September/2022	-	-	-	-	-	-	-	100%	-
October/2022	-	-	-	-	-	-	-	100%	-

* B.1.617.2 and parent lineages; ° B.1.1.529 and parent lineages.

## Data Availability

The sequencing data reported in this paper have been deposited in the GISAID portal (https://gisaid.org, accessed on 7 December 2023) under the GISAID Identifier: EPI_SET_231207zf, digital object identifier (DOI): https://doi.org/10.55876/gis8.231207zf, accessed on 7 December 2023 (Appendix A). The authors declare that all other data supporting the findings of this study are available within the article and its Appendix A files or are available from the corresponding author on reasonable request.

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
