# Peer review of "Assessing Genomic Mutations in SARS-CoV-2: Potential Resistance to Antiviral Drugs in Viral Populations from Untreated COVID-19 Patients"

_microorganisms, 2023, doi:10.3390/microorganisms12010002_

Round 1
Reviewer 1 Report
Comments and Suggestions for Authors
The study, "Assessing Genomic Mutations in SARS-CoV-2: Potential Resistance to Antiviral Drugs in Viral Populations from Untreated COVID-19 Patients," aims to understand the genetic diversity of RdRp complex subunits and Nsp5 in SARS-CoV-2 among untreated individuals in Southern Italy. The main contributions of this work include detailed analysis of specific mutations and their prevalence, offering critical insights into potential antiviral drug resistance. The paper's strengths lie in its methodological rigor, comprehensive data presentation, and relevance to current global health challenges.
Major Comments
Introduction – Contextual Depth (Lines 35-40): While the introduction sets the stage for the research, it would benefit from a deeper exploration of how viral mutations generally affect drug efficacy. This would provide a more comprehensive background for readers less familiar with virology.
Methods (Lines 156-158, Page 4): The methods section is thorough but lacks explicit mention of ethical considerations or approvals, especially concerning the use of human samples. It is crucial to address these aspects to maintain the ethical integrity of the research. For guidance, refer to the Declaration of Helsinki principles for ethical research involving human subjects. The authors states that ethical approval was not necessary due to the use of anonymized data from national surveillance. However, for enhanced clarity and to address potential concerns from readers, it would be beneficial to expand on this statement. Specifically, the authors could provide more details about the process of anonymization and the standards or guidelines followed to ensure data privacy and ethical compliance. This would reinforce the ethical rigor of the study and provide reassurance to the readers about the ethical handling of sensitive data.
Results – Cross-Variant Analysis (Lines 112-117, Page 5): The results section could be enhanced by including a discussion on the variability and significance of these mutations across different SARS-CoV-2 variants. This would offer a broader understanding of the mutations' impacts.
Minor Comments
Abstract (Lines 15-20): Consider adding more details in the abstract about the potential implications of the findings for antiviral drug resistance, which would offer a clearer understanding of the study's relevance.
Discussion (Lines 140-145, Page 7): The discussion would benefit from exploring how these mutations might impact future drug development or modification. This would add significant value to the paper by addressing broader implications.
Conclusions – Study Limitations (Lines 160-165, Page 8): In the conclusion, include a brief mention of the potential limitations of the study and suggestions for future research directions to provide a more comprehensive closure.
By addressing these major and minor comments, the paper could significantly enhance its academic rigor and broader impact. The inclusion of ethical considerations, a deeper exploration of the implications of mutations, and an expanded discussion on future research directions are particularly important for strengthening the study's contribution to the field.
Reviewer 2 Report
Comments and Suggestions for Authors
The authors Lombardo et al. in their manuscript titled “Assessing Genomic Mutations in SARS-CoV-2: Potential Resistance to Antiviral Drugs in Viral Populations from Untreated COVID-19 Patients” described the finding of many mutations in viral sequences of SARS-CoV-2 and in particular in some sequences affecting the areas of action of antivirals.
They describe an important analysis that correlates mutations in Nsp12, Nsp5, Nsp7, and Nsp8 sequences of the virus within a remarkable group of nasopharyngeal samples taken from SARS-CoV-2 positive patients during the three major waves of the COVID-19 Pandemic. The correlation with mutations affecting genomic areas where antivirals are supposed to take effect is a very important finding because it would open up new pharmacological scenarios.
These are very important and sensitive data and need clarifications and additions if the authors wish to make their data robust and clear.
The all-major revisions to be resolved are:
1. it needs to be indicated whether the subjects were vaccinated for COVID-19 or not. This information may be useful in understanding the trend of observed mutations.
2. A link needs to be included where the raw data are deposited so that due diligence can be performed.
3. It is also necessary to include DATA of mutations on the sequences that translate Spike protein and nucleocapsid protein.
4. Line 16: RdRp ?? RNA-dependent RNA polymerase--- Enter the first time the full name and as here so in the whole text for each abbreviation.
5. Line 16: you are not describing a virus population but a set of samples where sequences of a virus are extracted. Better replace the word viral population, it is not very scientific.
6. From line 17 to line 29, as well as in everything else when you first enter a virus region you must first write the whole name and then you can use the various abbreviations.
7. Line 28: “which could potentially lead to antiviral resistance in individuals who have not been treated with antiviral drugs”-- if samples from those who used antivirals have not been sequenced, it is incorrect to end the sentence as described. It is better to interrupt it in the middle: “which could potentially lead to antiviral resistance.”
8. The sentence from lines 50 to 52 is misspelled, please correct it.
9. The concept of where the replication mechanisms take place, i.e., in which cell (epithelial, other??) and anatomical district should be well included in the introduction considering that the authors are describing the basic concepts already reported by many authors on the topic of SARS-CoV-2
10. Line 93: SARS-CoV-2 populations??? Please change the wording. The term population is used for specimens and living species understood as capable of independent life not for viruses in general. Better to say: …for a group of samples with different sequences of SARS-CoV-2;
11. Sections 2.2 and 2.3 of materials and methods should be well-detailed, please include all the steps.
12. Line 130: indicate the channel used for BLASTn and enter the search reference (the link).
13. Line 130 to line 140: give the bibliographic references of the chosen setting and indicate the date of deposit of the sequences.
14. Line 149: briefly describe the Tamura-Nei method.
15. Lines 157: Were the consents collected from participants? It is not described. Clarify these aspects.
16. Line 210: replace letters aa with the word amino acids.
17. Please improve the quality of the figures.
18. Did the authors consider that the collected samples come from the epithelium-microbiome interface? They must recall studies in the literature that highlight the disruption of the surface microbiome. Some studies have also shown virus replication in bacterial cells and mutation resulting from these.
Round 2
Reviewer 2 Report
Comments and Suggestions for Authors
The authors have resolved most of the required items but 1 major revision and 1 minor revision remain.
I do not think it is ethically and scientifically rigorous and correct not to include the Spike and nucleocapsid protein mutations, especially because the authors have stated that they are one of the most relevant centers in Italy, one of the first countries to be affected by the pandemic, and especially because, if it is true what they show in the manuscript, that antivirals might not work on some viral mutations, so much should be true for other preventive solutions as well. Nevertheless, the study evaluates only the aspect of antivirals and affected sequences, and the authors' wishes are respected.
Major revision:
The authors must provide the GISAID platform access number as described in the guidelines. It is not possible yet to find the dates as it was written in the manuscript.
I have attached the instructions they must follow.
--Author(s) should use this section to communicate the data availability through the associated EPI_SET Identifier for all genome sequences and metadata used in the analysis.(https://gisaid.org/episet/)
Minor revision:
Not all abbreviations are written in full the first time are encountered in the text. They need to be resolved, ( ORF - open reading frame- line 48, and others in the text)
